# Induction of Apoptotic Temperature in Photothermal Therapy under Various Heating Conditions in Multi-Layered Skin Structure

**DOI:** 10.3390/ijms222011091

**Published:** 2021-10-14

**Authors:** Donghyuk Kim, Hyunjung Kim

**Affiliations:** Department of Mechanical Engineering, Ajou University, Suwon 16499, Gyeonggi-do, Korea; kimdonghyuk20@ajou.ac.kr

**Keywords:** apoptosis, heat transfer, photothermal therapy, squamous cell carcinoma, thermal damage

## Abstract

Recently, photothermal therapy has attracted attention as an alternative treatment to conventional surgical techniques because it does not lead to bleeding and patients quickly recover after treatment compared to incisional surgery. Photothermal therapy induces tumor cell death through an increase in the temperature using the photothermal effect, which converts light energy into thermal energy. This study was conducted to perform numerical analysis based on heat transfer to induce apoptosis of tumor tissue under various heating conditions in photothermal therapy. The Monte Carlo method was applied to evaluate a multi-layered skin structure containing squamous cell carcinoma. Tissue-equivalent phantom experiments verified the numerical model. Based on the effective apoptosis retention ratio, the numerical analysis results showed the quantitative correlation for the laser intensity, volume fraction of gold nanorods injected into the tumor, and cooling time. This study reveals optimal conditions for maximizing apoptosis within tumor tissue while minimizing thermal damage to surrounding tissues under various heating conditions. This approach may be useful as a standard treatment when performing photothermal therapy.

## 1. Introduction

The incidence of skin cancer is increasing each year because of the influence of global warming and increase in outdoor activities [1,2,3]. Skin cancer is divided into three types: squamous cell carcinoma, basal cell carcinoma, and malignant melanoma, which are generally treated by tumor incisions. However, treatment through incision is associated with recurrence due to incomplete incision of the tumor tissue; additionally, there are risks of bleeding occurs in the affected area and secondary infection [4,5,6,7]. Photothermal therapy has attracted attention as an alternative to incisional treatment to overcome these disadvantages [8,9,10]. Photothermal therapy uses the photothermal effect, in which light energy is converted into thermal energy to kill tumor tissue by increasing the temperature. This treatment is non-invasive compared to conventional treatment, reducing bleeding risks and increasing the rate of recovery [11,12].

Photothermal therapy supplies heat sources through various methods such as lasers and optic fibers. Among them, lasers in the near-infrared region are widely used because it is convenient to control the heat intensity and heating range [13,14,15]. However, using lasers in near-infrared regions results in less light absorption in normal and tumor tissue than in visible light regions [16]. A light absorber composed of a material that improves the light absorption of the medium is injected only into the tumor tissue to overcome this problem, enabling selective heating of the tumor tissue [17]. In this research field, gold nanoparticles, which have various advantages such as convenient surface modification and biosafety, are mainly used to enhance light absorption [18,19,20,21].

Various biological reactions, including death, depend on the temperature in typical biological tissues; the types of death mainly include apoptosis and necrosis [22,23]. In necrosis, metastasis and recurrence of cancer cells may occur through effects on surrounding tissues as cells die. Therefore, it is very important to induce apoptosis without affecting surrounding tissues. It is generally known that apoptosis occurs at temperatures between 43 °C and 50 °C, and necrosis occurs at temperatures above 50 °C [22,23]. Accordingly, in photothermal therapy, the apoptosis temperature range should be maintained by controlling the appropriate heat source, amount of gold nanoparticles injected, and total treatment time and prevent necrosis due to excessive temperature increases [24].

Numerous researchers have investigated the factors affecting photothermal therapy. Abo-Elfadl et al. [25] cultured human skin melanoma Sk-Mel-28 cells and square skin cell carcinoma in mice, injected gold nano semi-cubes into the tumor, and confirmed the tumor removal rate using photothermal therapy. The effect of photothermal therapy on partially inhibiting tumor growth and inflammation was confirmed. Maksimova et al. [26] demonstrated the effect of photothermal therapy by examining the temperature distribution and absorbed laser distribution when 808- and 810-nm lasers were irradiated to small animals injected with silica/gold nanoshell and malignant tumors through experiments and numerical analysis. This study revealed tumor tissue destruction when using various intensities and types of lasers for a fixed irradiation time, and confirmed that photothermal therapy effectively destroyed tumor tissues. Asadi et al. [27] presented the results of numerical analysis modeling based on magnetic resonance imaging for simulation and treatment planning of photothermal therapy using nanoparticles. This study utilized y-Fe203@Au nanoparticles, and the temperature and damage distribution for biological tissues was confirmed by the Pennes bioheat equation and Arrhenius damage model. After injecting nanoparticles into the CT26 cell line derived from mouse colon adenocarcinoma, the numerical analysis model was verified by irradiating a laser with a wavelength of 808 nm to evaluate the temperature distribution. Numerical and experimental results confirmed that damage was focused in the heating area and tumor tissue.

In summary, numerical analysis and experiments in these studies confirmed the light absorption and temperature distribution for limited conditions under the irradiated laser, showing that tumor tissue was destroyed. However, the apoptosis rate in the tumor tissue and thermal damage to the surrounding normal tissues were not quantified by simply checking the presence of damage using the Arrhenius damage integral model or just phenomenologically confirming the removal rate of the tumor through experiments. In addition, photothermal therapy could not be performed under various conditions, such as an extended laser irradiation time, and the conditions for achieving the optimal therapeutic effect were not determined.

Therefore, in this study, the temperature distribution of the tumor tissue and surrounding multi-layered skin structure for different laser irradiation times and various laser intensities, as well as the volume fraction of the injected gold nanorods (GNR), was confirmed through numerical analysis. In addition, by applying the apoptotic variable proposed by Kim et al. [28], the apoptosis rate in the tumor tissue and amount of thermal damage to the surrounding normal tissue were quantified to determine the appropriate irradiation time and laser intensity and volume fraction of the injected GNR to achieve optimal therapeutic effects.

## 2. Materials and Methods

### 2.1. Monte Carlo Method and Heat Transfer Model

When lasers represented by the heat source of photothermal therapy used to irradiate biological tissue, scattering occurs simultaneously as increasing amounts of heat from the laser are absorbed within the medium. The Monte Carlo method is used in bioheat transfer to analyze this behavior [29]. This method is a probabilistic calculation of the movement of laser photons through random numbers when a laser irradiates the medium.

In the Monte Carlo method, the distance and direction in which a photon moves are determined by a random number. First, the distance a photon moves is determined as in Equations (1) and (2) using the total attenuation coefficient of the medium and a randomly selected number, where *S* is the distance moved by a photon per one time; *ξ* is a random number generated between 0 and 1; and $\mu_{a}$, $\mu_{s}$, $\mu_{tot}$ are the light absorption coefficient, light scattering coefficient, and total attenuation coefficient of the medium, respectively:(1)$$S=\frac{-\ln(\xi )}{\mu_{tot}}$$
(2)$$\mu_{tot}=\mu_{a}+\mu_{s}$$

For the angle of movement of photons, the deflection angle and azimuth are calculated as shown in Equations (3) and (4) using the anisotropic factor, a variable that determines the directionality in which particles are scattered, where cos*θ* is the deflection angle, *ψ* is the azimuth, and *g* is the anisotropy factor:(3)$$\cos\theta =\{\begin{array}{cc} \frac{1}{2g}\left\{1+g^{2}-{\left[\frac{1-g^{2}}{1-g+2g\xi }\right]}^{2}\right\} & if\;g>0\\ 2\xi -1 & if\;g=0 \end{array}$$
(4)ψ=2πξ

Once the azimuth and deflection angle have been determined, the direction vector in the Cartesian coordinate system can be calculated using Equations (5)–(7), where $\mu_{x}$, $\;\mu_{y}$, $\;\mu_{z}$ are the direction cosines for each axis:(5)$$\mu_{x}^{\prime }=\frac{\sin\theta }{\sqrt{1-\mu_{z}^{2}}}(\mu_{x}\mu_{z}\cos\psi -\mu_{y}\sin\psi )+\mu_{x}\cos\theta$$
(6)$$\mu_{y}^{\prime }=\frac{\sin\theta }{\sqrt{1-\mu_{z}^{2}}}(\mu_{y}\mu_{z}\cos\psi -\mu_{x}\sin\psi )+\mu_{z}\cos\theta$$
(7)$$\mu_{z}^{\prime }=-\sin\theta \cos\psi \sqrt{1-\mu_{z}^{2}}+\mu_{z}\cos\theta$$
(8)$$\Delta W=W\frac{\mu_{a}}{\mu_{tot}}$$

Finally, when the distance and angle at which the photon moves are determined, the energy reduction of the photon according to one movement of the photon is ascertained from the optical properties of the medium, as shown in Equation (8), and moves until the energy converges to zero through energy reduction due to absorption from the medium. *W* is the energy weight of the photon. The overall simulation flow chart is shown in Figure 1.

Once the final movement path of a photon is determined by the Monte Carlo method mentioned above, the absorption distribution of laser heat in the medium is determined according to the intensity of the irradiated laser per unit area and optical properties of the medium. The temperature distribution of the medium over time can thus be calculated through the thermal diffusion equation of Equation (9), where *q* is the amount of heat absorbed by the medium, *k* is the thermal conductivity, *ρ* is the density, and $c_{v}$ is the specific heat:(9)$$\frac{\partial T}{\partial \tau }=\frac{q+\nabla \cdot (k\nabla T)}{\rho c_{v}}$$
(10)$$\begin{array}{l} \Delta T=\frac{\Delta \tau }{\rho c_{v}}(\mu_{a}FP_{l}dxdydz+(T_{x^{-}}-T)\frac{2kk_{x^{-}}}{k+k_{x^{-}}}\frac{dydz}{dx}+(T_{x^{+}}-T)\frac{2kk_{x^{+}}}{k+k_{x^{+}}}\frac{dydz}{dx}+(T_{y^{-}}-T)\frac{2kk_{y^{-}}}{k+k_{y^{-}}}\frac{dxdz}{dy}\\ +(T_{y^{+}}-T)\frac{2kk_{y^{+}}}{k+k_{y^{+}}}\frac{dxdz}{dy}+(T_{z^{-}}-T)\frac{2kk_{z^{-}}}{k+k_{z^{-}}}\frac{dxdy}{dz}+(T_{z+}-T)\frac{2kk_{z+}}{k+k_{z^{+}}}\frac{dxdy}{dz}) \end{array}$$

In this study, the thermal diffusion equation of Equation (9) was calculated using the explicit finite element method (Equation (10)), where *F* is the fluence rate, *P_l_* is the intensity of the laser, and *dx*, *dy*, *dz* are the differential lengths of each axis [30].

### 2.2. Apoptotic Variable

The final goal of photothermal therapy is to maximize apoptosis in tumor tissue when the laser is irradiated on tumor tissue while minimizing thermal damage to surrounding normal tissue. Three apoptotic variables proposed by Kim et al. [28] were used to confirm this effect quantitatively.

First, the apoptosis ratio ($\theta_{A}$), which confirms the volume ratio corresponding to the apoptosis temperature range in the tumor tissue between 43 °C and 50 °C, is expressed as a ratio of the volume corresponding to the apoptosis temperature to the total tumor volume as shown in Equation (11). For example, if all parts of the tumor fall within the apoptosis temperature range, $\theta_{A}$ has a value of 1. This study aims to determine the effects of treatment time on photothermal therapy. However, as $\theta_{A}$ confirms the rate at which apoptosis occurs within the tumor at a specific time to determine how long the tumor tissue maintains the temperature band corresponding to apoptosis during the total treatment time, as shown in Equation (12), the average for the total treatment time was used. This value was named as the apoptosis retention ratio ($\theta_{A}^{*}$), where *τ* represents the total treatment time:(11)$$\theta_{A}=\frac{Apoptosis\;volume\;(if\;43<V(T)<50)\;}{Tumor\;volume}$$
(12)$$\theta_{A}^{*}=\frac{1}{\tau }\int_{0}^{\tau }\theta_{A}(\tau )d\tau$$

Second, the thermal hazard value ($\theta_{H,n}$), which represents the degree of thermal damage to normal tissues, is determined by weighting each of the phenomena in biological tissues according to temperature, as shown in Table 1. The variable can be calculated as a ratio of the weighted sum of volumes belonging to each temperature band to the total volume of the surrounding normal tissue, as shown in Equation (13). If there is no thermal damage to the surrounding normal tissue, $\theta_{H,n}$ becomes 1; if thermal damage increases as the temperature increases, $\theta_{H,n}$ has a value of 1 or more. As this variable also identifies results at a specific time, thermal damage in the surrounding normal tissues was identified by averaging the $\theta_{H,n}$ at each time to obtain results over the total treatment time, as shown in Equation (14). This value was named as the thermal hazard retention value ($\theta_{H,n}^{*}$):(13)$$\theta_{H,n}=\frac{\sum_{j=1}^{m}\left(V_{n}(T)\cdot w_{j}\right)}{V_{n}}$$
(14)$$\theta_{H,n}^{*}=\frac{1}{\tau }\int_{0}^{\tau }\theta_{H,n}(\tau )d\tau$$

However, as shown in Equation (13), $\theta_{H,n}$ is calculated as a ratio to the volume of the surrounding normal tissue. Accordingly, it is necessary to establish a standard area of surrounding normal tissue that should be viewed. We confirmed the thermal damage to normal tissues from the end of the tumor to 50% of the diameter of the tumor tissue [31].

Finally, the effective apoptosis retention ratio ($\theta_{A,eff}^{*}$) to confirm the final goal of photothermal therapy, which is to maximize apoptosis in tumor tissue while minimizing thermal damage to the surrounding normal tissue, is a ratio between $\theta_{A}^{*}$ and $\theta_{H,n}^{*}$, as shown in Equation (15). Thus, various conditions that produce the optimal treatment effect can be determined by simultaneously confirming the degree to which the apoptosis temperature is maintained according to the total treatment time and degree of maintaining thermal damage to surrounding normal tissues:(15)$$\theta_{A,eff}^{*}=\frac{Apoptosis\;retention\;ratio\;(\theta_{A}^{*})}{Thermal\;hazard\;retention\;value\;of\;normal\;tissue\;(\theta_{H,n}^{*})}$$

### 2.3. Experiment for Validation of the Numerical Model

In skin cancer, which was evaluated this study, it remains difficult to experimentally determine the treatment effect because the ability to analyze patients is limited. Therefore, a phantom with similar thermal properties to that of the human body is manufactured and used for experiments in this research field. In this study, an acrylamide-based tissue-equivalent phantom proposed by Surowiec et al. [33] and Iizuka et al. [34] was used. The phantom exhibits thermal properties similar to those of human skeletal muscle tissue. The thermal properties of the skeletal muscle tissue and phantom are shown in Table 2.

The tissue-equivalent phantom used in the experiment was produced with an acrylamide stock solution followed by addition of a catalyst and coagulant. The mixture of each ingredient for producing the phantom is shown in Table 3.

In this study, to simulate GNR injection, a simulating tumor tissue part with a diameter of 10 mm and depth of 10 mm was manufactured. GNR were injected in a volume fraction of $2\times 10^{-5}$ and surrounded with a phantom without GNR with 40 mm diameter and 30 mm length to simulate normal tissue. where the volume fraction represents the volume of gold nanoparticles relative to the volume of the tumor and is a dimensionless number. Figure 2 shows information on the produced phantom. The red circle in Figure 2a indicates the phantom part that mimics the tumor tissue and has a bright purple color because of the effect of GNR. In this study, GNR with a diameter of 10 nm and length of 67 nm was used. Figure 2b shows a schematic diagram of the manufactured phantom and location of the thermocouple attached to measure temperature. The temperature was measured after attaching a T-type thermocouple at a depth of 1 mm from the surface and at 4 positions along the radial direction. Temperatures were recorded by a data acquisition system (34972A, Agilent Technologies, Santa Clara, CA, USA) and PC.

Figure 3 is an experimental device for the numerical analysis verification experiment. The experiment was conducted on an optic table to prevent interference from external vibrations. A wavelength laser of 1064 nm with a diameter of 1 mm was used as a heat source. The laser diameter was increased to 10 mm by a beam expander, and the laser path was changed from horizontal to vertical using an optical mirror. Finally, a laser with the same diameter as the phantom mimicking the tumor tissue was irradiated vertically to heat the phantom. The resulting temperature change in the radial direction of the phantom was confirmed.

Figure 4 shows the temperature measurement results of experiments and numerical analyses at various locations over time. The experiment was conducted for 20 min after setting the initial temperature to 20 °C, and the temperature difference between the initial temperature and corresponding time point was confirmed. As shown in Figure 4, the average RMSE between the experiment and numerical analysis at various points was derived as 0.1677. Based on these results, the numerical analysis model used in this study is suitable.

### 2.4. Numerical Investigation

The human skin is divided into four layers, which is considered in a numerical analysis model. Figure 5 shows a cross-sectional view of the numerical model. A cylindrical tumor with a diameter and length of 10 and 3.5 mm, respectively, was in normal tissue of a three-dimensional cuboid structure with a width, length, and depth of 30, 30, and 10 mm, respectively, and gold nanorods were assumed to be uniformly distributed within the tumor. The tumor was at a depth of 0.1 mm from the surface and a laser with a wavelength of 1064 nm with an appropriate penetration depth into the tumor tissue was used as the heat source [37]. The thickness and thermal optical properties of each skin layer and tumor are shown in Table 4.

When a laser irradiates tumor tissue, including GNR, a photothermal effect is generated because of light absorption in the GNR, causing the temperature of tumor tissue to increase. This heat is diffused to surrounding normal tissues through conduction, as shown in Figure 6a. If the laser is irradiated with a reasonable cooling time, the conducted heat does not spread widely, as shown in Figure 6b because of the thermal confinement effect [46]. Heat transfer occurs within a narrow range. Accordingly, in this study, cooling time conditions were added to numerical analysis to confirm the temperature distribution of the medium according to various cooling times.

Table 5 shows the numerical analysis conditions to confirm photothermal therapy in various situations. The laser diameter was set to 10 mm, which is equal to the tumor diameter, and the laser profile was set to be top-hat. A previous study [31] confirmed that the temperature distribution in the medium converged when the laser was irradiated to the tumor tissue for more than 1000 s. Therefore, the laser irradiation time, which is the total treatment time, was selected as 120–960 s in 120 s intervals. The laser intensity was 0–2000 mW, and the volume fraction of the injected GNR was divided into four stages from 10^−3^ to 10^−6^ at intervals of 10^−1^ to confirm the temperature distribution under the given conditions. Finally, the heating and cooling times were set to 15, 20, 30, and 60 s.

The optical properties of GNR were determined using the effective light absorption area (*r_eff_*), absorption efficiency (*Q*), and volume fraction (*f_v_*) of GNR in the tumor tissue, as shown in Equations (16) and (17) [47]. In this study, GNRs with lengths and diameters of 67 and 10 nm, which are known to have the highest absorption efficiency for lasers with a wavelength of 1064 nm, were used. The absorption efficiency *Q* of the GNR was calculated using the discrete dipole approximation method [48]:(16)$$\mu_{a,np}=0.75f_{v}\frac{Q_{a}\;}{r_{eff}},\;\;\;\mu_{s,np}=0.75f_{v}\frac{Q_{s}\;}{r_{eff}}$$
(17)$$r_{eff}=\sqrt[{3}]{\frac{3V}{4\pi }}$$
(18)$$\mu_{a}=\mu_{a,m}+\mu_{a,np},\;\;\;\mu_{s}=\mu_{s,m}+\mu_{s,np}$$

Finally, the optical properties of the tumor tissue injected with GNR can be calculated from the sum of the optical properties of the tumor tissue and GNR, as shown in Equation (18) [49]. Table 6 shows the optical properties of the entire tumor tissue for the various volume fractions of the injected GNR.

## 3. Results and Discussion

### 3.1. Temperature of Tumor and Normal Tissue under Various Conditions

Figure 7 shows temperature changes in tumor and normal tissues for different volume fractions (*f_v_*) of GNR with a laser intensity of 500 mW, total treatment time of 360 s, and heating and cooling times (*τ_h_ & τ_c_*) of 30 s. In order to confirm the temperature change inside the tumor tissue and surrounding normal tissue, the temperature change was confirmed at a depth of 2 mm, the central part of the tumor, and at a depth of 4 mm, a part of the normal tissue adjacent to the tumor. When *f_v_* was 10^−3^, 10^−4^**,** and 10^−5^, both tumor and normal tissues showed a similar tendency; when *f_v_* was 10^−6^, the tumor tissue rose at a lower temperature range because of lower light absorption. Figure 7a shows the temperature change at the central position of the tumor tissue with a depth of 2 mm from the surface, and the green shaded area indicates 43–50 °C, which is known to cause apoptosis. When *f_v_* is 10^−6^, this corresponds to a temperature range when apoptosis occurs between 65 and 190 s. Additionally, as shown in Figure 7b, normal tissue death began at approximately 70 s after starting treatment, with the thermal damage intensified when around 50 °C was reached after 360 s. Based on these results, the effects of photothermal therapy over the treatment time were quantified under various conditions by calculating the apoptosis retention ratio ($\theta_{A}^{*}$) and thermal hazard retention value ($\theta_{H,n}^{*}$) over time.

Figure 8 shows the temperature change of tumor and normal tissues according to different *τ_h_ & τ_c_*. When the laser intensity was 500 mW, the total treatment time was 360 s and *f_v_* was 10^−6^. The temperature of the tumor tissue showed a smaller increase when no cooling time was included compared to when a cooling time was implemented because of the thermal confinement effect. In addition, compared to a cooling time of 60 s, a time of 20 s corresponds more to the temperature range at which apoptosis occurs in the total treatment time. Hence, the cooling time giving the optimal therapeutic effect was confirmed.

### 3.2. Apoptosis Retention Ratio

As described above, photothermal therapy is a treatment technique that increases the temperature of the tumor tissue and kills the tissue by using a heat source represented by a laser. As it is important to maintain the temperature range corresponding to apoptosis between 43 °C and 50 °C as much as possible, the temperature distribution within the tumor tissue must be quantitatively confirmed to determine the time rate at which apoptosis occurs. Accordingly, we identified the apoptosis retention ratio ($\theta_{A}^{*}$) for different laser intensities and different volume fractions (*f_v_*) of GNR in tumors for each heating and cooling time (*τ_h_ & τ_c_*).

Figure 9 shows a graph of $\theta_{A}^{*}$ according to the laser intensity and *f_v_* for various *τ_h_ & τ_c_* when the total treatment time was 360 s. The $\theta_{A}^{*}$ showed a maximum value when *f_v_* was 10^−6^ for all *τ_h_ & τ_c_*. Additionally, the intensity of the laser with the maximum value of $\theta_{A}^{*}$ was determined, and we showed that the laser intensity from which the maximum value of $\theta_{A}^{*}$ was derived increased as *f_v_* decreased. This is because, as the *f_v_* decreases, light absorption at the tumor tissue decreases, and more energy is required to increase the temperature to the range corresponding to apoptosis. This result confirms that a specific *f_v_* and laser intensity have an optimal treatment effect for various *f_v_*.

Figure 10 shows a graph of $\theta_{A}^{*}$ according to *τ_h_ & τ_c_* and laser intensity for various treatment times when *f_v_* was 10^−6^. For each treatment time, the *τ_h_ & τ_c_* and laser intensity showed optimal treatment effects. For example, when the total treatment time was 360 s when *τ_h_ & τ_c_* were 60 s each, the apoptosis temperature range was reached in more parts of the tumor compared to at other times. Furthermore, as the total treatment time increased, the laser intensity with the maximum value of $\theta_{A}^{*}$ decreased. This is because increasing the total treatment time increases the laser irradiation time, which increases the amount of heat transferred to the tumor tissue and results in an excessive temperature rise in the tumor. Therefore, to maintain the apoptosis temperature range, the laser intensity must be lowered.

### 3.3. Thermal Hazard Retention Value of Normal Tissue

When the laser irradiates tumor tissue, light energy is converted into heat energy by the photothermal effect, and the temperature of the tumor tissue increases. Heat transfer occurs by conduction to the surrounding normal tissue. Thus, if the tumor tissue reaches a temperature resulting in apoptosis (between 43 °C and 50 °C) by adjusting the appropriate laser intensity and amount of injected GNR, the temperature of the surrounding normal tissue is 43 °C or higher because of heat transfer from the tumor tissue. This may cause thermal damage. Therefore, the thermal damage of normal tissue from the end of tumor tissue to 50% of the diameter of tumor tissue was quantitatively confirmed using the thermal hazard retention value ($\theta_{H,n}^{*}$).

Figure 11 shows a graph of $\theta_{H,n}^{*}$ according to the laser intensity and volume fraction of GNR (*f_v_*) in the tumor for various heating and cooling times (*τ_h_ & τ_c_*) when the total treatment time was 960 s. As the laser intensity increased, thermal damage to surrounding normal tissues increased. This is because, at a higher laser intensity, more heat is absorbed from the tumor tissue and more heat transfer to the surrounding normal tissue occurs. In addition, when *f_v_* was 10^−6^, thermal damage was lower than at other volume fractions of GNR. This is because a smaller *f_v_* leads to a smaller amount of heat being absorbed by the tumor tissue. As a result, the amount of heat transferred to the surrounding normal tissue was decreased.

Figure 12 shows a graph of $\theta_{H,n}^{*}$ according to the laser intensity and heating and cooling times (*τ_h_ & τ_c_*) for various total treatment times when *f_v_* was 10^−6^. We confirmed that $\theta_{H,n}^{*}$ increased as *τ_h_ & τ_c_* increased. This is because the cumulative *τ_c_* is the same over the total treatment time, but a shorter *τ_c_* correlates with a shorter irradiation time, thus reducing the heating time and preventing the high temperature from being reached immediately. Furthermore, as the total treatment time increased, the value of $\theta_{H,n}^{*}$ increased, as described in Section 3.2. As the total treatment time increased, the amount of heat absorbed by the tumor increased, resulting in higher conduction to surrounding normal tissues.

### 3.4. Effective Apoptosis Retention Ratio

As described above, the ultimate goal of photothermal therapy is to maximize apoptosis of tumor tissue while minimizing thermal damage to surrounding normal tissue. Accordingly, in this study, the optimal treatment conditions were identified using the effective apoptosis retention ratio ($\theta_{A,eff}^{*}$), which can simultaneously confirm the apoptosis retention ratio ($\theta_{A}^{*}$) and thermal hazard retention value ($\theta_{H,n}^{*}$). As it was confirmed from the results in Section 3.2 and Section 3.3 that the maximum value of $\theta_{A}^{*}$ and minimum value of $\theta_{H,n}^{*}$ were derived when the volume fraction of GNR (*f_v_*) in the tumor was 10^−6^, the final $\theta_{A,eff}^{*}$ confirmed the result when the *f_v_* was 10^−6^.

Figure 13 shows a contour graph for $\theta_{A,eff}^{*}$ according to the total treatment time (*τ_tot_*), laser intensity (*P_l_*), and heating and cooling times (*τ_h_ & τ_c_*) when *f_v_* was 10^−6^. As *τ_tot_* increased, the *P_l_* showing the optimal treatment effect decreased. This is because, similar to the previous trend for $\theta_{A}^{*}$, as the total treatment time increased, the amount of heat absorbed by the tumor tissue increased, resulting in an excessive temperature rise in the tumor and surrounding normal tissues. This enables determination of the *P_l_* that derives the optimal treatment effect according to the total treatment time. For various *τ_h_ & τ_c_*, better treatment conditions depend on each treatment time.

Finally, Table 7 summarizes the *P_l_*, *τ_h_ & τ_c_*, and $\theta_{A,eff}^{*}$ under the conditions showing the optimal treatment effect for each treatment time. The *τ_h_ & τ_c_* and *P_l_* showed the optimal therapeutic effect for each treatment time. Finally, among the numerical analysis conditions evaluated in this study, when the treatment time was 960 s, the treatment effect was strongest under the conditions of 350 mW of *P_l_* and 20 s of *τ_h_ & τ_c_*, respectively. This makes it possible to provide information on the conditions used to derive the optimal therapeutic effect when actually performing photothermal therapy for skin cancer.

## 4. Conclusions

In this study, numerical modeling of actual skin composed of four layers containing squamous cell carcinoma was performed, and quantitative information on various conditions was obtained by using the Monte Carlo method, which inferred the optimal photothermal treatment effect for squamous cell carcinoma near the skin layer based on the effective apoptosis retention ratio.

The numerical modeling results were verified in acrylamide-based phantom experiments. The distribution of light absorption in the medium was derived using the Monte Carlo method that considers the scattering and absorption of the irradiated laser simultaneously. Based on this, the thermal diffusion equation obtained the temperature distribution of the tumor tissue and surrounding normal tissue. Tumor tissue apoptosis and the amount of thermal damage to surrounding normal tissue were quantified for various total treatment times, heating and cooling times, volume fraction of injected GNR, and laser intensity in photothermal therapy.

By calculating the effective apoptosis retention ratio, which is the ratio of the apoptosis retention ratio to the thermal hazard retention value, was used to determine a condition that minimizes thermal damage to surrounding normal tissues and maximizes the occurrence of apoptosis in the tumor tissue, which is the purpose of photothermal therapy. This confirmed the conditions for the volume fraction of GNR in the tumor, the laser intensity, and the heating and cooling time that derive the optimal treatment effect at various treatment times. Through this, this study can be used as a basis for optimal treatment conditions in photothermal therapy. These results should be validated in in vivo experiments under the determined conditions.

## Figures and Tables

**Figure 1 ijms-22-11091-f001:**
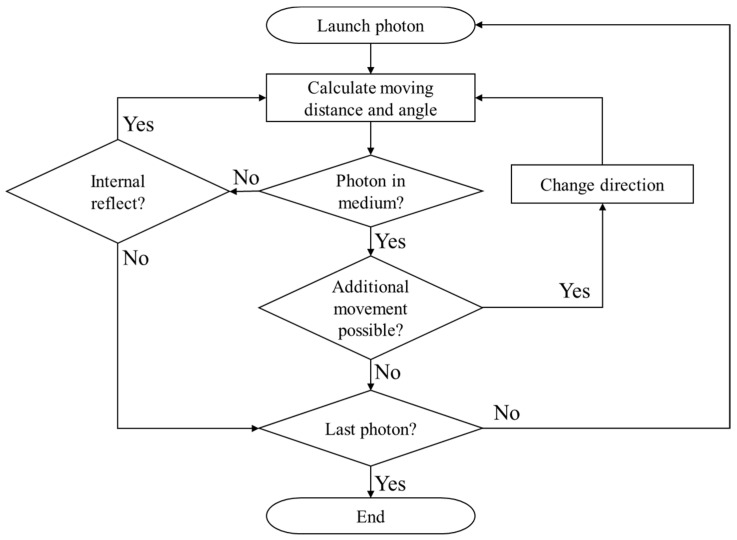
Flow chart of the Monte Carlo method.

**Figure 2 ijms-22-11091-f002:**
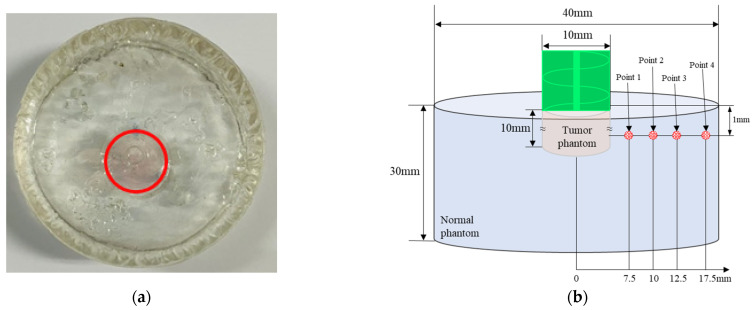
(**a**) Photograph of phantom (red line: tumor phantom boundary); (**b**) geometry of phantom and location of inserted thermocouples.

**Figure 3 ijms-22-11091-f003:**
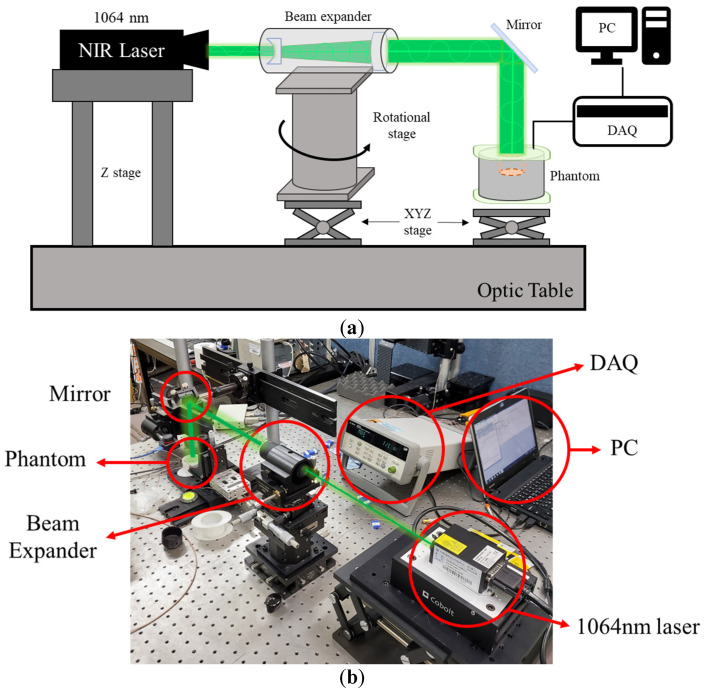
(**a**) Schematic of experiment apparatus; (**b**) photograph of experiment apparatus.

**Figure 4 ijms-22-11091-f004:**
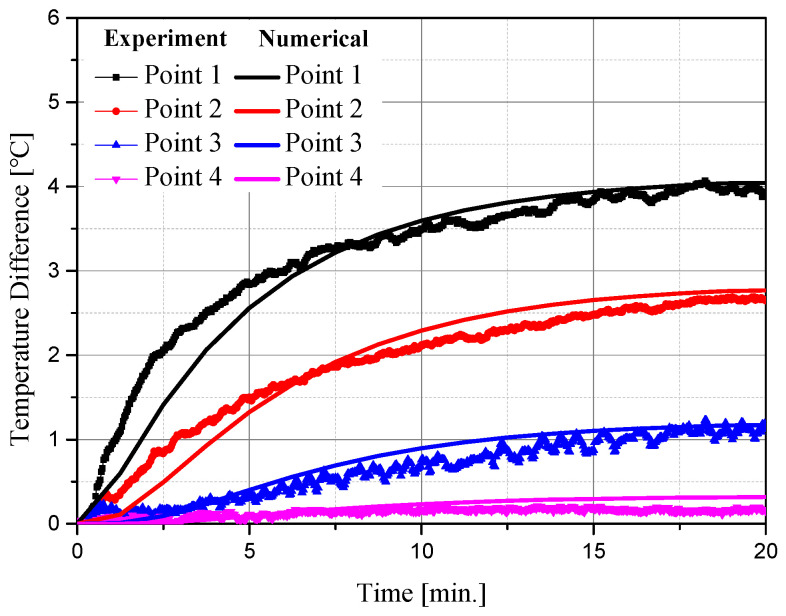
Comparison of temperature measurement results of the experiment and numerical analysis.

**Figure 5 ijms-22-11091-f005:**
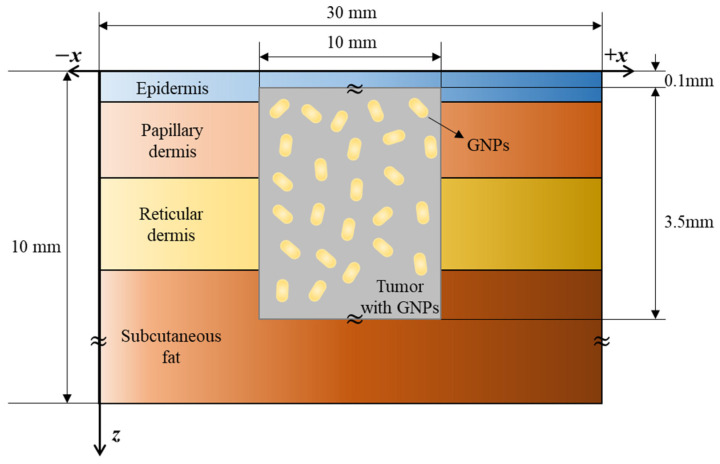
Schematic of numerical model.

**Figure 6 ijms-22-11091-f006:**
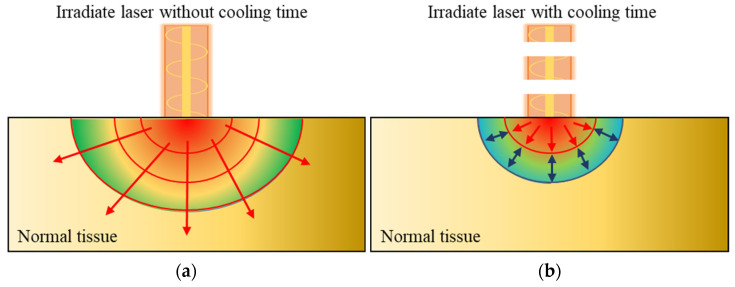
(**a**) Heat transfer in normal tissue; (**b**) thermal confinement effect by applying cooling time.

**Figure 7 ijms-22-11091-f007:**
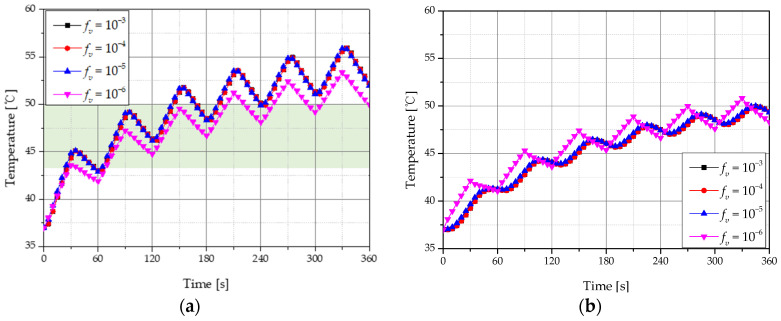
Temperature change with 30 s heating and cooling time for various volume fractions of GNR (**a**) tumor tissue (depth: 2 mm); (**b**) normal tissue (depth: 4 mm).

**Figure 8 ijms-22-11091-f008:**
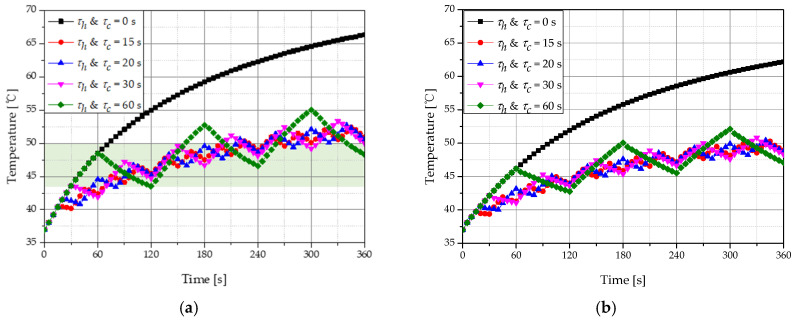
Temperature change for various heating and cooling time (**a**) tumor tissue (depth: 2 mm); (**b**) normal tissue (depth: 4 mm).

**Figure 9 ijms-22-11091-f009:**
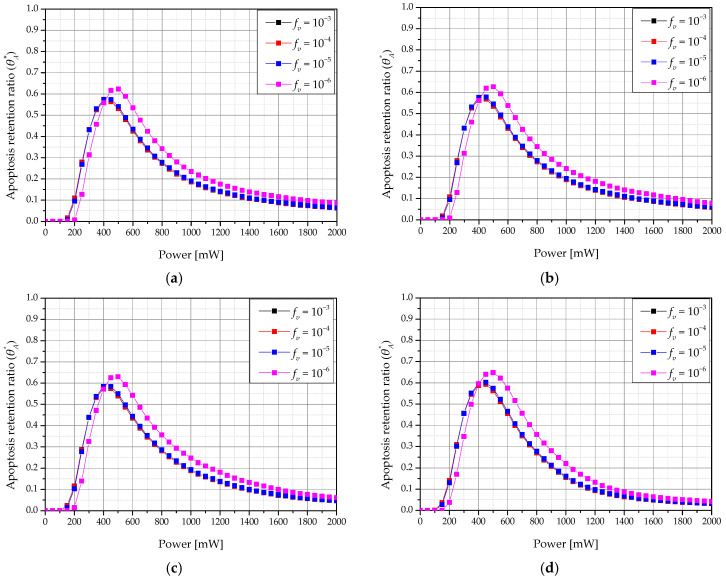
Apoptosis retention ratio ($\theta_{A}^{*}$) for various volume fraction of GNR (*f_v_*) (*τ_tot_* = 360 s) (**a**) *τ_h_ & τ_c_* = 15 s; (**b**) *τ_h_ & τ_c_* = 20 s; (**c**) *τ_h_ & τ_c_* = 30 s; (**d**) *τ_h_ & τ_c_* = 60 s.

**Figure 10 ijms-22-11091-f010:**
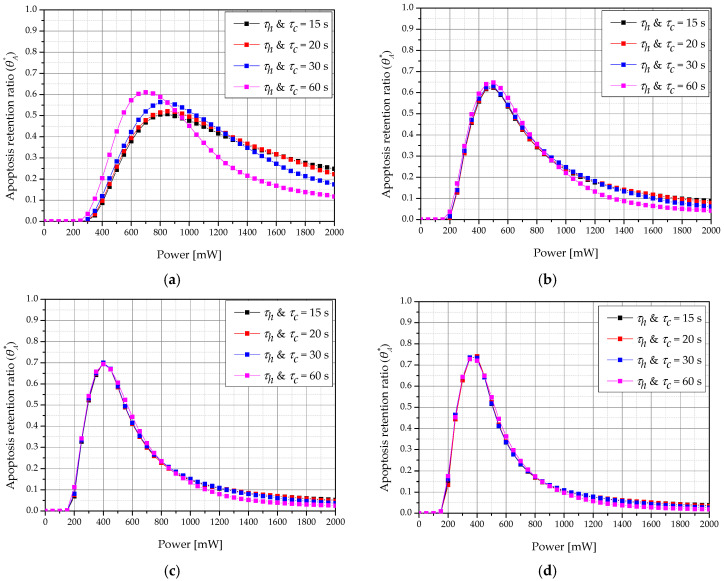
Apoptosis retention ratio ($\theta_{A}^{*}$) for various heating/cooling time (*τ_h_ & τ_c_*) (*f_v_* = 10^−6^) (**a**) *τ_tot_* = 120 s; (**b**) *τ_tot_* = 360 s; (**c**) *τ_tot_* = 600 s; (**d**) *τ_tot_* = 840 s.

**Figure 11 ijms-22-11091-f011:**
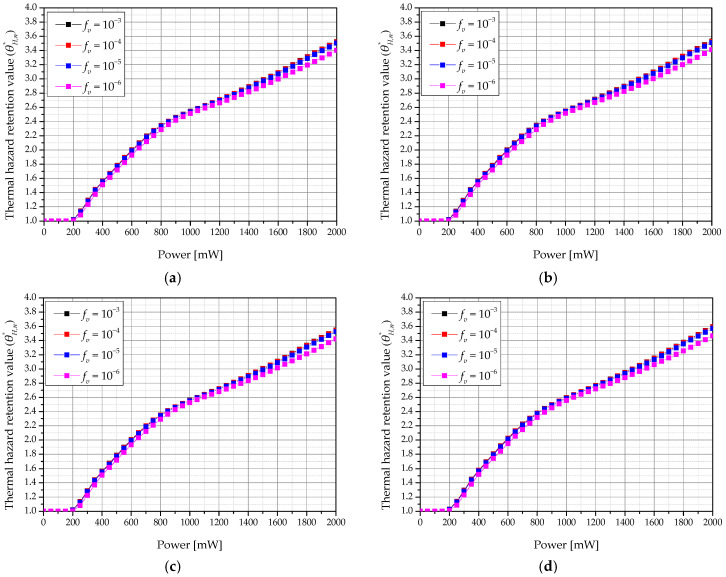
Thermal hazard retention value of normal tissue ($\theta_{H,n}^{*}$) for various volume fraction of GNR (*f_v_*) (*τ_tot_* = 960 s). (**a**) *τ_h_ & τ_c_* = 15 s; (**b**) *τ_h_ & τ_c_* = 20 s; (**c**) *τ_h_ & τ_c_* = 30 s; (**d**) *τ_h_ & τ_c_* = 60 s.

**Figure 12 ijms-22-11091-f012:**
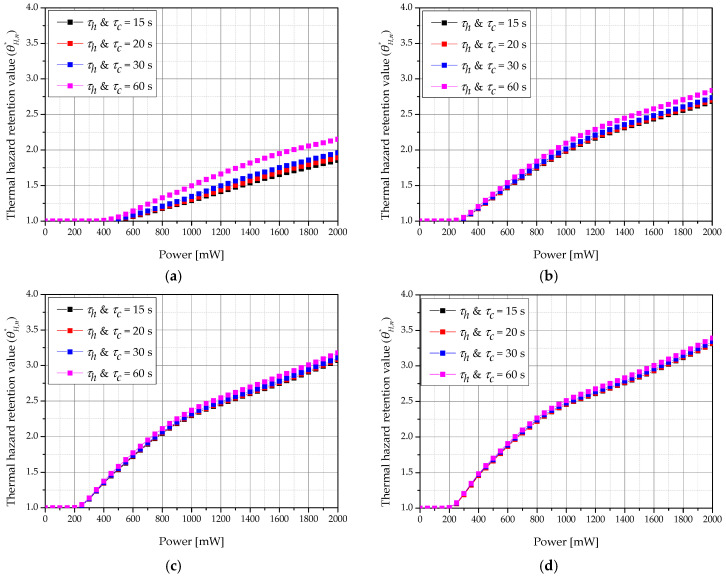
Thermal hazard retention value of normal tissue ($\theta_{H,n}^{*}$) for various heating/cooling time (*τ_h_ and τ_c_*) (*f_v_* = 10^−6^). (**a**) *τ_tot_* = 120 s; (**b**) *τ_tot_* = 360 s; (**c**) *τ_tot_* = 600 s; (**d**) *τ_tot_* = 840 s.

**Figure 13 ijms-22-11091-f013:**
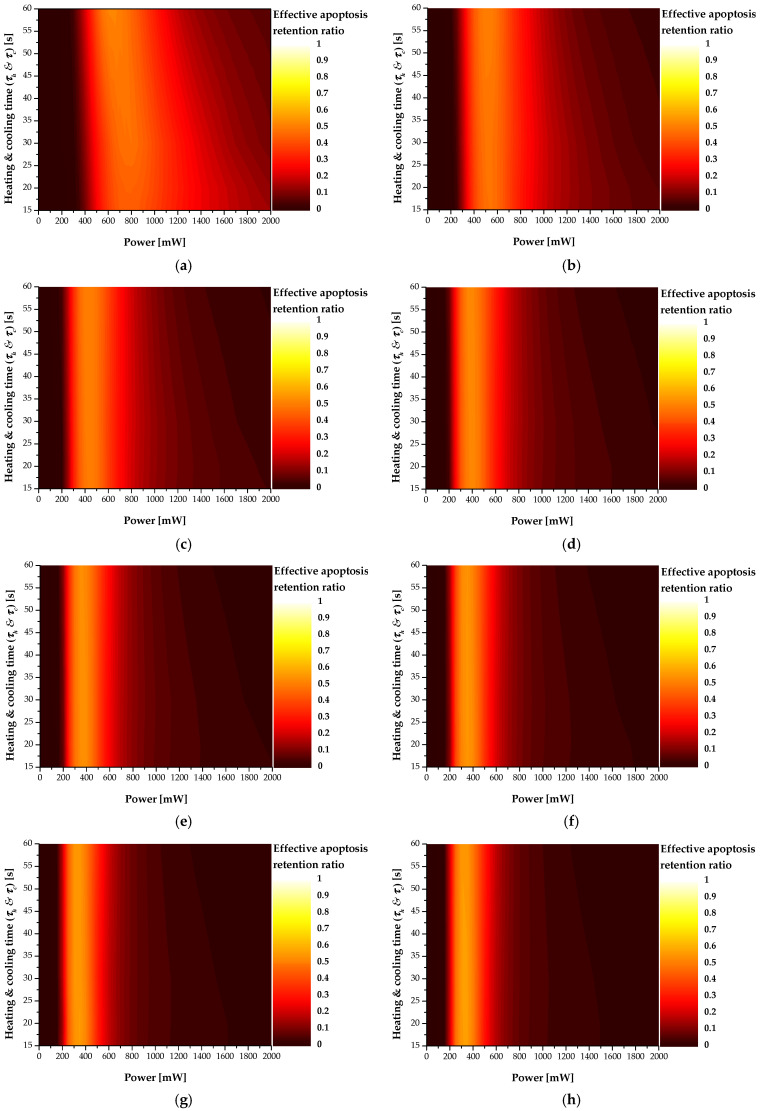
Effective apoptosis retention ratio ($\theta_{A,eff}^{*}$) for various total treatment time (*τ_tot_*) (*f_v_* = 10^−6^) (**a**) *τ_tot_* = 120 s; (**b**) *τ_tot_* = 240 s; (**c**) *τ_tot_* = 360 s; (**d**) *τ_tot_* = 480 s; (**e**) *τ_tot_* = 600 s; (**f**) *τ_tot_* = 720 s; (**g**) *τ_tot_* = 840 s; (**h**) *τ_tot_* = 960 s.

**Table 1 ijms-22-11091-t001:** Laser-induced thermal effects [28,32].

Temperature Range (°C)	Biological Effect	Weight, w
37	Normal	1
37<T<43	Biostimulation	1
43≤T<45	Hyperthermia	2
45≤T<50	Reduction in enzyme activity	2
50≤T<70	Protein denaturation (coagulation)	3
70≤T<80	Welding	4
80≤T<100	Permeabilization of cell membranes	5
100≤T<150	Vaporization	6
150≤T<300	Carbonization	7
T>300	Rapid cutting and ablation	8

**Table 2 ijms-22-11091-t002:** Thermal properties for tissue-equivalent phantom and human muscle.

	Polyacrylamide Phantom	Skeletal Muscle Tissue [35]
Density (kg/m^3^)	1070	1070
Specific heat (J/kgK)	3810	3470
Thermal conductivity (W/mK)	0.56 [36]	0.535

**Table 3 ijms-22-11091-t003:** Recipe of polyacrylamide phantom.

Materials	Stock (% by Weight)	Remarks
Acrylamide	26.0	Acrylamide stock solution
N, N’-methylenebisacrylamide	0.2	
Sodium chloride	1.05	
De-ionized water	71.7	
Ammonium persulphate (10%) (APS)	1	coagulant
*N*,*N*,*N*′,*N*′-Tetramethylethylenediamine (TEMED)	0.5	catalyst

**Table 4 ijms-22-11091-t004:** Depth and thermal and optical properties of the skin layers and tumor [16,38,39,40,41,42,43,44,45].

	t (mm)	ρ (kg/m^3^)	$c_{p}$ (J/kgK)	k (W/mK)	$\mu_{a}$ (1/mm)	$\mu_{s}$ (1/mm)	g
Epidermis	0.08	1200	3589	0.235	0.4	45	0.8
Papillary dermis	0.5	1200	3300	0.445	0.38	30	0.9
Reticular dermis	0.6	1200	3300	0.445	0.48	25	0.8
Subcutaneous fat	7.82	1000	2500	0.19	0.43	5	0.75
Tumor	3.5	1070	3421	0.495	0.08	1.28	0.925

**Table 5 ijms-22-11091-t005:** Geometry of simulation conditions.

Numerical Parameter	Case	Number	Remarks
treatment time (*τ_tot_*)	120 to 960 s	8	Intv: 120 s
Laser power (*P_l_*)	0 to 2000 mW	21	Intv: 50 mW
Volume fraction of GNPs (*f_v_*)	10^−3^ to 10^−6^	4	Intv: 10^−1^
Heating and cooling time (*τ_h_/τ_c_*)	15, 20, 30, 60	4	

**Table 6 ijms-22-11091-t006:** Optical properties of tumor with GNR for various volume fractions.

Volume Fraction of GNR	10^−3^	10^−4^	10^−5^	10^−6^
Absorption coefficient ($\mu_{a}$) (mm^−1^)	118.419	11.914	1.263	0.198
Scattering coefficient ($\mu_{s}$) (mm^−1^)	6.101	1.762	1.328	1.285

**Table 7 ijms-22-11091-t007:** Best treatment effect condition and effective apoptosis retention ratio for various treatment time.

Treatment Time (s)	Optimal Treatment Conditions		
	*τ_h_ & τ_c_* (s)	*P_l_* (mW)	$\theta_{A,eff}^{*}$
120	60/60	650	0.50736
240	60/60	500	0.49713
360	30/30	450	0.49589
480	20/20	400	0.51446
600	30/30	350	0.52502
720	30/30	350	0.54085
840	20/20	350	0.55117
960	20/20	350	0.55429

## Data Availability

Data sharing is not applicable to this article.

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
