# Peer review of "Induction of Apoptotic Temperature in Photothermal Therapy under Various Heating Conditions in Multi-Layered Skin Structure"

_ijms, 2021, doi:10.3390/ijms222011091_

Round 1

Reviewer 1 Report

Dear editor,

I appreciate the article but is looks alike the previous study (https://doi.org/10.3390/app11031103) Please check it (imges, text, information, similar sentences....) . If you don't have issues with that, it can be considerer for publication. 

Reviewer 2 Report

The submitted manuscript has discussed the theoretical study of photothermal cancer therapy. The NIR irradiated a mathematical model meaningfully presented GNR responsive photothermal heat generation. Overall the study is unique, as most of the studies in this area focused only on preclinical exploration. It would be an interesting addition to photothermal cancer therapy. However, few questions need to address; it is including:

  1. Is there any relation between using a 1064 nm wavelength laser and 40 nm GNR? Why have the authors chosen the 1064 nm wavelength laser?
  2. In Figure 2a, there is a red circle. It has to describe in the figure caption.
  3. Do the authors administrate GNR inside the phantom? If they did, it has to mention in the experimental section with the concentration of GNR.
  4. There is a Figure (a) before Figure 3; is it part of Figure 2? It is not very clear.
  5. In Figure 7, the temperature of tumor and normal tissue were studied 2 mm and 4 mm, respectively. Is there any reason for selecting a different depth?
  6. The conclusion section needs to be concise. It would be better to use less than four paragraphs.

Round 2

Reviewer 1 Report

I agree with the manuscript final version for publication.

Author Response

Thanks for your kindness response.

Reviewer 2 Report

[1] The concentration unit of GNR should mention in manuscript. Does the volume fraction of 2×10−5 mean number of GNRs or mole unit?

[2] The Figure 3 (a) and (b) need to organize in same row.
